# Effects of Ca(NO₃)₂ Stress on Mitochondria and Nitrogen Metabolism in Roots of Cucumber Seedlings

**Yang Yang** [1,2,†], **Zhiyuan Lu** [1,2,†], **Jie Li** [1,2], **Ling Tang** [1,2], **Shaoke Jia** [1,2], **Xuming Feng** [1,2,3], **Chenggang Wang** [1,2,3], **Lingyun Yuan** [1,2,3], **Jinfeng Hou** [1,2,3] **and Shidong Zhu** [1,2,3,*]

1 College of Horticulture, Vegetable Genetics and Breeding Laboratory, Anhui Agricultural University, Hefei 230036, China; yangyang1028@ahau.edu.cn (Y.Y.); 17720091@ahau.edu.cn (Z.L.); jieli@ahau.edu.cn (J.L.); tl17730228448@163.com (L.T.); jsk19930714@163.com (S.J.); xumingfeng@ahau.edu.cn (X.F.); cgwang@ahu.edu.cn (C.W.); ylyun99@163.com (L.Y.); houjinfeng@ahau.edu.cn (J.H.)
2 Provincial Engineering Laboratory for Horticultural Crop Breeding of Anhui, Hefei 230036, China
3 Department of Vegetable Culture and Breeding, Wanjiang Vegetable Industrial Technology Institute, Maanshan 238200, China
* Correspondence: sdzhuaau@sina.cn; Tel./Fax: +86-0551-6578-6212
† These authors contributed equally to this work.

**Abstract:** Cucumber is a major vegetable crop grown in facilities, and its seedlings are sensitive to salinity. Thus, it is important for cucumber cultivation to research the effects of Ca(NO₃)₂. In this study, we investigated salt-sensitive Chunqiu (CQ) and salt-tolerant BoMei 49 (BM) seedlings, the changes to the enzymes involved in the mitochondria antioxidant system in the seedlings, and the changes in the MPTP (Mitochondrial Permeability Transition Pore) opening, mitochondrial membrane fluidity, mitochondrial membrane potential depolarization, mitochondrial electron transfer chain, and NO metabolism in response to Ca(NO₃)₂. Ca(NO₃)₂ stresses inhibited the growth of the cucumber seedlings, which showed a reduced root length, a decreased surface area and a decreased number of root tips, but a significantly increased root diameter. The mitochondrial malondialdehyde (MDA) content, $H_2O_2$ content, and $O_2^-$ producing rate of the two cucumber roots were gradually increased in the Ca(NO₃)₂ treatment. The activity of superoxide dismutase (SOD) and peroxidation enzyme (POD) gradually increased, and catalase (CAT) activity gradually decreased. The electron transport chain activity of "BM" was higher than "CQ" under Ca(NO₃)₂ stress. Ca(NO₃)₂ stress destroyed the membrane structure of the mitochondria, so that the mitochondrial body tended to bend, causing double-membrane digestion and a hollow interior. Under the stress of Ca(NO₃)₂, the $NO_3^-$ content of the seedlings increased significantly. The contents of $NH_4^+$ and NO, as well as the activities of nitrate reductase (NR), nitrite reductase (NIR), and nitric oxide synthase (NOS) decreased significantly. These findings provide physiological insights into root damage in response to salt stress for salt tolerance in cucumber.

**Keywords:** cucumber; Ca(NO₃)₂ stress; mitochondria; nitrogen metabolism

## 1. Introduction

Soil salinization is one of the most important environmental factors that may limit the crop growth of facilities [1]. Soil salinization is mainly caused by excessive nitrate and calcium ions in the soil due to improper fertilization and other causes [2]. Changes in osmotic stress and cell structure are induced by Ca(NO₃)₂. Therefore, it leads to increased cellular levels of $Ca^{2+}$ and $NO_3^-$ that inhibit

plant growth nutritional absorption, especially moisture and ion absorption [3]. $Ca^{2+}$ is the second messenger in the plant. Studies using ion elemental analysis and X-ray microanalysis have shown that 80 mM $Ca(NO_3)_2$ has an effect on the uptake and distribution of mineral nutrients in cucumber seedlings [4]. It not only regulates $Na^+/K^+$ but also regulates various antioxidant and NO pathways to the plant. $Ca(NO_3)_2$ stresses increase oxidative damage due to osmotic stress and structure damage, which can seriously damage plant cells. Under the stress of $Ca(NO_3)_2$, the ion balance and energy supply of the cucumber roots were broken, and the growth of the plant roots was inhibited. Some studies have shown that calcium nitrate promotes the accumulation of photosynthetic pigments in leaves and affects the photosynthesis of plants [1].

Reactive oxygen species (ROS) are highly destructive and can cause membrane lipid peroxidation, organelle damage, and enzyme inhibition [5], all of which contribute to cell death. To remove ROS and maintain redox homeostasis, plants have developed a series of antioxidant defense systems to prevent oxidative damage caused by high levels of ROS. These include superoxide dismutase (SOD), catalase (CAT), peroxidation enzyme (POD), and so on. ROS is mainly produced in the mitochondria while attacking the mitochondrial membrane structure, which has a great impact on mitochondrial function. However, the electron transport chain in mitochondria participates in NO production process when producing ROS. Previous studies have shown that various external stresses from the environment caused oxidative damage and functional problems in mitochondria, such as the mitochondrial tricarboxylic acid (TCA) cycle [6], electron transport chain complexes [7], and substrate phosphorylation [8]. Limited information is currently available in the literature regarding the effect of different $Ca(NO_3)_2$ levels on overall plant root mitochondria and NO metabolism in mitochondria. Cucumber is a major vegetable crop grown in facilities, and its seedlings are sensitive to salinity. Thus, it is important for cucumber cultivation to research the effects of $Ca(NO_3)_2$. In this study, we investigated cucumber seedlings, the changes from enzymes involved in the mitochondria antioxidant system in the seedlings, and the changes in the mitochondrial electron transfer chain and NO metabolism in response to different $Ca(NO_3)_2$ levels.

## 2. Materials and Methods

### 2.1. Plant Materials and Growth Conditions

Cucumber (*Cucumis sativus* L.) seeds ("Chunqiu" marked "CQ", China, Beijing, Beijing Flower-goddess Agriculture Co.,Ltd, salt-sensitive; "BoMei 49" marked "BM", China, Tianjin, Tianjin Derit Seed Co.,Ltd, salt-tolerant) were disinfected with 55 °C warm water for 15 min, then washed thoroughly with de-ionised water (Table S1). They were germinated in filter papers in petri dishes in the dark for approximately 24 h at 29 °C. When the seed radicle broke through the seed coat by about 2 mm, they were transferred into plastic nursery trays (50 cm × 50 cm × 5 cm) containing fine sand [9]. The seedling temperature was controlled at 29 ± 2 °C in the day and 19 ± 2 °C in the night. After cotyledon expansion, seedings were cultivated with half-strength Hoagland nutrient solution; after true leaf expansion, seedings were planted with Hoagland nutrient solution. At the second-leaf stage, seedlings of uniform size were transferred to a crate [10] that contained Hoagland nutrient solution and was aerated with an air pump at an interval of 20 min to maintain the dissolved oxygen (DO) level at $8.0 ± 0.2$ mg $L^{-1}$. Fifteen cucumber seedlings were planted in one crate. The seedlings were cultivated in a greenhouse. The highest temperature during the day was 31 °C, and the lowest temperature at night was 23 °C; the relative humidity was 51–71% during their growth period under natural illumination [11]. After one day of the transplanting seedling stage, seedlings were treated with 70 Mm (Table S2) $Ca(NO_3)_2$. Samples (roots) were taken at 10:00 after 0, 3, 6, 9, and 12 days, and the physiological indexes of each treatment were measured by mixed sampling.

The root of the treated seedlings was sampled 0, 3, 6, 9, and 12 days after treatment. Mitochondria were immediately extracted and frozen in liquid nitrogen and stored at −80 °C for further experiments.

## 2.2. Analyses of Root Morphology and Mitochondria Extraction

Three seedlings were randomly selected from each replicating group. They were first rinsed with distilled water and then were analyzed using the root system analyzer with WinRHIZO pro 2013 (Regent Instruments Inc., Canada). Mitochondria were isolated by differential centrifugation followed by purification in a discontinuous precool gradient as described by Kuhn et al. [12].

## 2.3. Ultrastructure of Root Tips Cell

The ultrastructure of the root cell was analyzed using electron micrographs. The root tips were cut to pieces of approximately 1 mm$^2$. The plant material was fixed with 3% glutaraldehyde and post-fixed with 1% osmium tetroxide ($O_sO_4$). After dehydrating in acetone, they were embedded in Durcupan ACM Fluka (Solarbio, Tongzhou district, Beijing, China). After polymerization of the resin, the resulting root fraction was cut to obtain ultra-thin sections, stained with uranium acetate and lead citrate in series, and examined using a transmission electron microscope H-7650 (Hitachi, Tokyo, Japan) at an accelerating voltage of 80 kV [13].

## 2.4. Determination of Mitochondrial Lipid Peroxidation, Mitochondrial Free Radical Production, Mitochondrial $H_2O_2$ Level, and Antioxidant Enzyme Activity

Lipid peroxidation was determined using thiobarbituric acid methods as described by Heath et al. [14]. $O_2^-$ generation rates were measured following the method of Rauckman [15]. The $H_2O_2$ level was estimated by using kits from Nanjing Jiancheng company. SOD activity was assayed based on the method of Giannopolitis and Ries [16]. One unit of SOD activity was defined as the amount of enzyme that was required to cause a 50% inhibition of the reduction of NBT as monitored at 560 nm. POD activity was measured by the increase in absorbance at 470 nm due to the oxidation of guaiacol [17]. CAT activity in the mitochondria was assayed spectrophotometrically according to Dhindsa et al. [18].

## 2.5. Determination of Mitochondrial Enzyme Activities and Mitochondrial Membrane Properties

The enzyme activities of four mitochondrial complexes were estimated by using Solarbio (Tongzhou district, Beijing, China) kits (the article numbers were BC0515, BC3235, BC3245, and BC0945, respectively, Beijing, China). Mitochondrial membrane potential was measured according to the method described by Yuan [19]. The changes in the absorbance were determined to understand the opening of the mitochondrial permeability transition pore (MPTP) following the methods of Wang [20]. Membrane fluidity was inferred using fluorescence polarization anisotropy. Membrane suspensions were incubated with the fluorescent probe 8-aniline-1-naphthalenesulfonic acid (ANS) at room temperature for 1 min. Measurements were made at room temperature using a Hitachi RF-5400 spectrofluorometer (Hitachi, Tokyo, Japan). The excitation and the emission wavelengths were 400 nm and 480 nm [21].

## 2.6. Determination of $NH_4^+$, $NO_3^-$ Concentrations, and Enzyme Activity Involved in N Assimilation

Nitrate and ammonium nitrogen determination in roots was performed as described by Meng [22]. NR and NIR activities were assayed by the method of Liu [23]. The content of NO and NOS activity was determined using kits from the Nanjing Jiancheng company (Nanjing, Jiangsu province, China).

## 2.7. Statistical Analysis

The data were statistically analyzed using two-way ANOVA analysis of variance and Duncan's multiple range tests with a $p < 0.05$ indicating significance. Typically, three replicates were used per treatment.



## 3. Results

### 3.1. Root Morphology

The root length and tips in seedlings of the two cultivars decreased from 0 days to 12 days after $Ca(NO_3)_2$ treatment, and the decrease was most significant at 12 days. However, the root diameter of the two varieties increased with the number of days after treatment. After 12 days of the $Ca(NO_3)_2$ stress treatment, the average diameter of "CQ" and "BM" increased significantly, and the growth trend of the two varieties was similar. The variation trend of the root length, mean diameter, and root tip number of the cucumbers increased with the increase in stress time, while the variation range of the surface area was relatively small (Table 1).

**Table 1.** Effect on root morphological structure 0, 3, 6, 9, and 12 days after treatment with $Ca(NO_3)_2$ stress. The *p* values marked with similar letters are not significantly different (*p* < 0.05). All the values are the mean of three replicates ± standard deviation (SD).

| | Length (cm) | | Surface Area (cm$^2$) | | Average Diam (mm) | | Tips | |
| | Chunqiu (CQ) | BoMei 49 (BM) | CQ | BM | CQ | BM | CQ | BM |
|---|---|---|---|---|---|---|---|---|
| 0 days | 546.63 ± 12.67b | 609.30 ± 12.24a | 148.59 ± 3.18cd | 203.86 ± 1.45a | 0.8219 ± 0.0096c | 0.9222 ± 0.0295de | 3586.67 ± 124.23b | 4064.00 ± 75.50a |
| 3 days | 405.86 ± 23.50cd | 522.01 ±46.02b | 153.25 ± 2.66c | 184.51 ± 6.75b | 1.1292 ± 0.0677cd | 1.1423 ± 0.0750cd | 1609.00 ±153.62d | 2301.00 ± 6.08c |
| 6 days | 392.25 ± 16.65cd | 500.26 ± 4.02b | 153.22 ± 6.64c | 154.22 ± 4.63c | 1.1630 ± 0.0474cd | 1.5013 ± 0.0943b | 1756.33 ± 86.78d | 1676.67 ± 33.46d |
| 9 days | 346.63 ± 2.22d | 413.50 ± 15.05c | 157.53 ± 7.83c | 138.55 ± 2.30d | 1.4002 ± 0.0309bc | 1.5480 ± 0.0636b | 902.00 ± 43.66e | 1000.00 ± 44.8e |
| 12 days | 185.10 ±10.03e | 234.11 ±21.55e | 85.35 ± 6.27e | 136.79 ± 4.47d | 1.4844 ± 0.1791b | 2.2367 ± 0.1365a | 794.33 ± 18.187e | 849.33 ± 69.29e |

### 3.2. Content of Lipid Peroxides, Free Radical Production, and Antioxidant Enzyme Activity in Mitochondria

$Ca(NO_3)_2$ stress caused a significant increase in the production rate of $H_2O_2$, malondialdehyde (MDA), and $O_2^-$ in seedling roots of CQ and BM (Table 2). Six days after treatment, the MDA content under $Ca(NO_3)_2$ treatment was significantly higher in BM, but the increase in CQ was higher than that of BM; the increased trend also showed in BM after the 6th day but was not significant. The change trend of free radical production was similar to that of MDA content. This indicates that under $Ca(NO_3)_2$ stress, MDA content and free radical production in the mitochondria of salt-sensitive cucumber root is higher than that of salt-tolerant cucumber (Table 2).

**Table 2.** Effect of $Ca(NO_3)_2$ stress on malondialdehyde (MDA), hydrogen peroxide($H_2O_2$), and the $O_2$-generation rate in cucumber root mitochondria. The *p* values marked with similar letters are not significantly different (DMRT (Duncan's multiple range test), *p* < 0.05). All the values are the mean of three replicates ± SD.

| | MDA (µmol/g Protein) | | $H_2O_2$ (µmol/g Protein) | | $O_2$-Generation Rate (µmol/g·min$^{-1}$ Protein) | |
| | CQ | BM | CQ | BM | CQ | BM |
|---|---|---|---|---|---|---|
| 0 days | 0.7368 ± 0.0171g | 0.6480 ± 0.01671h | 51.33 ± 1.034i | 55.65 ± 0.81i | 0.2209 ± 0.0075d | 0.1472 ± 0.0089e |
| 3 days | 1.9701 ± 0.0050e | 1.7591 ± 0.03220f | 107.51 ± 0.55f | 73.25 ± 1.05h | 0.2736 ± 0.0018c | 0.2452 ± 0.0029d |
| 6 days | 2.3557 ± 0.0465c | 2.2407 ± 0.01835d | 139.71 ± 3.54d | 90.46 ± 0.27g | 0.3296 ± 0.0074b | 0.2851 ± 0.0234c |
| 9 days | 3.1500 ± 0.0064b | 2.2382 ± 0.0110d | 170.87 ± 3.11b | 115.81 ± 1.15e | 0.4952 ± 0.0058a | 0.3318 ± 0.0036b |
| 12 days | 3.5217 ± 0.03362a | 2.2667 ± 0.0239d | 191.77 ± 5.69a | 150.13 ± 0.91c | 0.5059 ± 0.0029a | 0.3200 ± 0.0030ab |

Under $Ca(NO_3)_2$ stresses, SOD activities and POD activities in mitochondria of cucumber seedling roots were significantly increased. The increasing trend found for SOD activity was the same for the two varieties, and the SOD activity of the two varieties was similar in each period. The changing POD trend of "BM" is the same as that of "CQ", but the overall value is higher than that of "CQ". CAT decreased significantly after $Ca(NO_3)_2$ treatment. At 6 days of stress treatment, "CQ" and "BM" decreased more than the other treatment times (Figure 1).

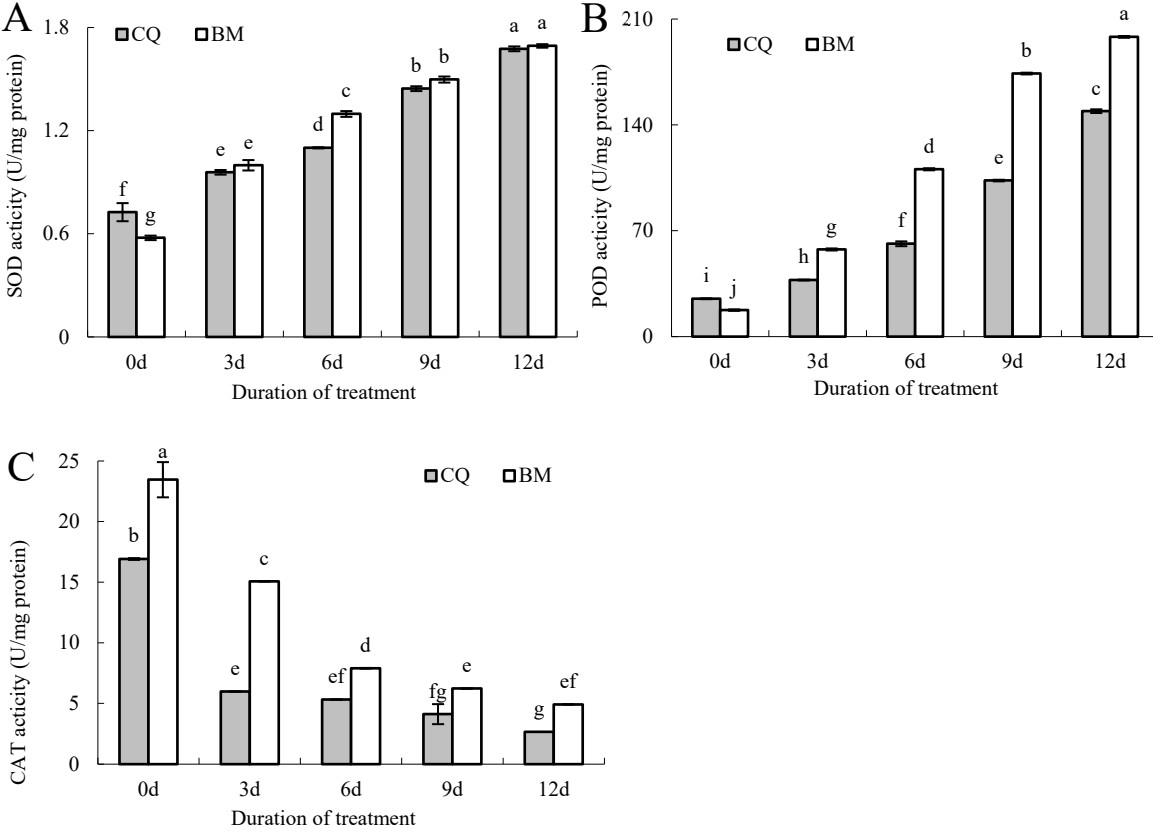

**Figure 1.** Determination of the activities of SOD (**A**), POD (**B**), and CAT (**C**) in the mitochondria of cucumber roots at 0, 3, 6, 9, and 12 days after Ca(NO$_3$)$_2$ treatment. Values are the means of three independent experiments. Bars marked with dissimilar letters are significantly different according to Duncan's multiple range tests ($p < 0.05$).

### 3.3. Ca(NO$_3$)$_2$ Effects on Electron Transport Chain in Mitochondria

The activities of complex I (NADH/ubiquinone oxidoreductase enzymes) and complex III (ubiquinol/cytochrome c reductase) were reduced in the root mitochondria (Figure 2). The trend of complex II activity was different from that of complex I. With the increase of Ca(NO$_3$)$_2$ stress time, the activity of "CQ" complex II continued to decrease, while the activity of "BM" complex II gradually increased. Complex IV (cytochrome c oxidase) activity of the two varieties was different from the above trend. Complex IV activity was greatly increased 3 days after treatment and then it was significantly decreased 6 days after treatment.

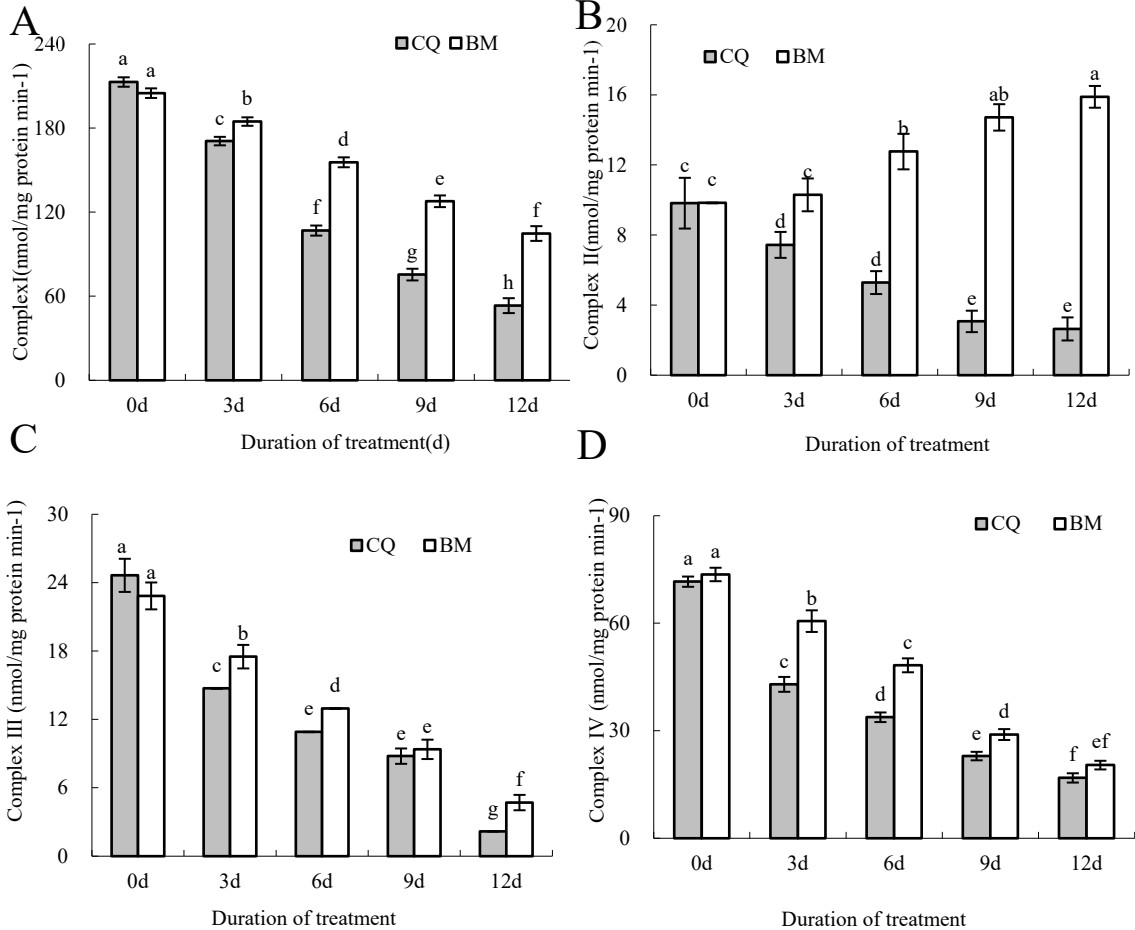

**Figure 2.** Determination of the activities of complex I—NADH/ubiquinone oxidoreductase (**A**), complex II—succinate dehydrogenase (**B**), complex III—ubiquinol/cytochrome c reductase (**C**), and complex IV—cytochrome c oxidase (**D**) in the mitochondria of cucumber roots at 0, 3, 6, 9, and 12 days after $Ca(NO_3)_2$ treatment. Values are the means of three independent experiments. Bars marked with dissimilar letters are significantly different according to Duncan's multiple range tests ($p < 0.05$).

*3.4. MPTP Opening, Membrane Fluidity, Membrane Potential, and ATP Level*

To evaluate the roles of mitochondria, MPTP opening, membrane fluidity, and the mitochondrial membrane potential depolarization were further examined. Table 3 shows that root mitochondria MPTP opening of two varieties increased significantly from 0 days to 3 days. MPTP opening had no significant change after 3 days ($p < 0.05$); meanwhile, it was significantly higher in BM than in CQ cucumber from 6 days to 12 days. Two varieties membrane fluidity had no difference under treatment (Figure 3b). In the two varieties, membrane fluidity had no difference under the treatment. However, it decreased slightly from $Ca(NO_3)_2$ stress. Twelve days after stress, membrane fluidity was 26% higher in BM than in CQ. As shown in Table 3, the mitochondrial membrane potential of the two varieties after 12 days decreased dramatically ($p < 0.05$). Moreover, mitochondrial membrane potential for CQ declined by 39.6%, 43.6%, 52.1%, and 59.1% after 3, 6, 9, and 12 days, respectively. BM also shows the same trend but was lower than CQ. Mitochondrial ATP levels in the CQ root decreased significantly by 71.1% under $Ca(NO_3)_2$ stress at 12 days, while it decreased to 19.4% in BM compared with the 0-day treatment (Figure 4).

**Table 3.** Effect of $Ca(NO_3)_2$ stress on Cytochrome c/a, MPTP opening, membrane fluidity, and membrane potential in cucumber root mitochondria. The *p* values marked with similar letters are not significantly different (DMRT, $p < 0.05$). All the values are the mean of three replicates ± SD.

| | Cytochrome c/a | | MPTP Opening | | Membrane Fluidity | | Membrane Potential | |
|---|---|---|---|---|---|---|---|---|
| | CQ | BM | CQ | BM | CQ | BM | CQ | BM |
| 0 days | 1.1854 ± 0.0038a | 1.1455 ± 0.0065a | 0.2213 ± 0.0127c | 0.2467 ± 0.0107d | 39.65 ± 0.18a | 41.15 ± 0.12a | 20.96 ± 0.10a | 25.09 ± 0.34a |
| 3 days | 1.0091 ± 0.0002b | 1.0367 ± 0.0094b | 0.7040 ± 0.0069a | 0.5287 ± 0.0068c | 26.93 ± 0.03b | 33.13 ± 1.12b | 12.65 ± 0.52b | 18.01 ± 0.58b |
| 6 days | 0.9979 ± 0.0030c | 1.0171 ± 0.0044c | 0.6733 ± 0.0176ab | 0.5487 ± 0.0192bc | 22.36 ± 0.36c | 30.27 ± 0.35c | 11.82 ± 0.12c | 16.27 ± 0.83c |
| 9 days | 0.9639 ± 0.0082c | 0.9979 ± 0.0007d | 0.6627 ± 0.0013b | 0.5873 ± 0.0064a | 20.87 ± 0.30d | 28.63 ± 0.30cd | 10.02 ± 0.08d | 14.74 ± 0.27c |
| 12 days | 0.9180 ± 0.0090d | 0.9493 ± 0.0038e | 0.6567 ± 0.0033b | 0.5787 ± 0.0047ab | 20.24 ± 0.23d | 27.53 ± 0.75d | 8.55 ± 0.05e | 12.85 ± 0.07d |

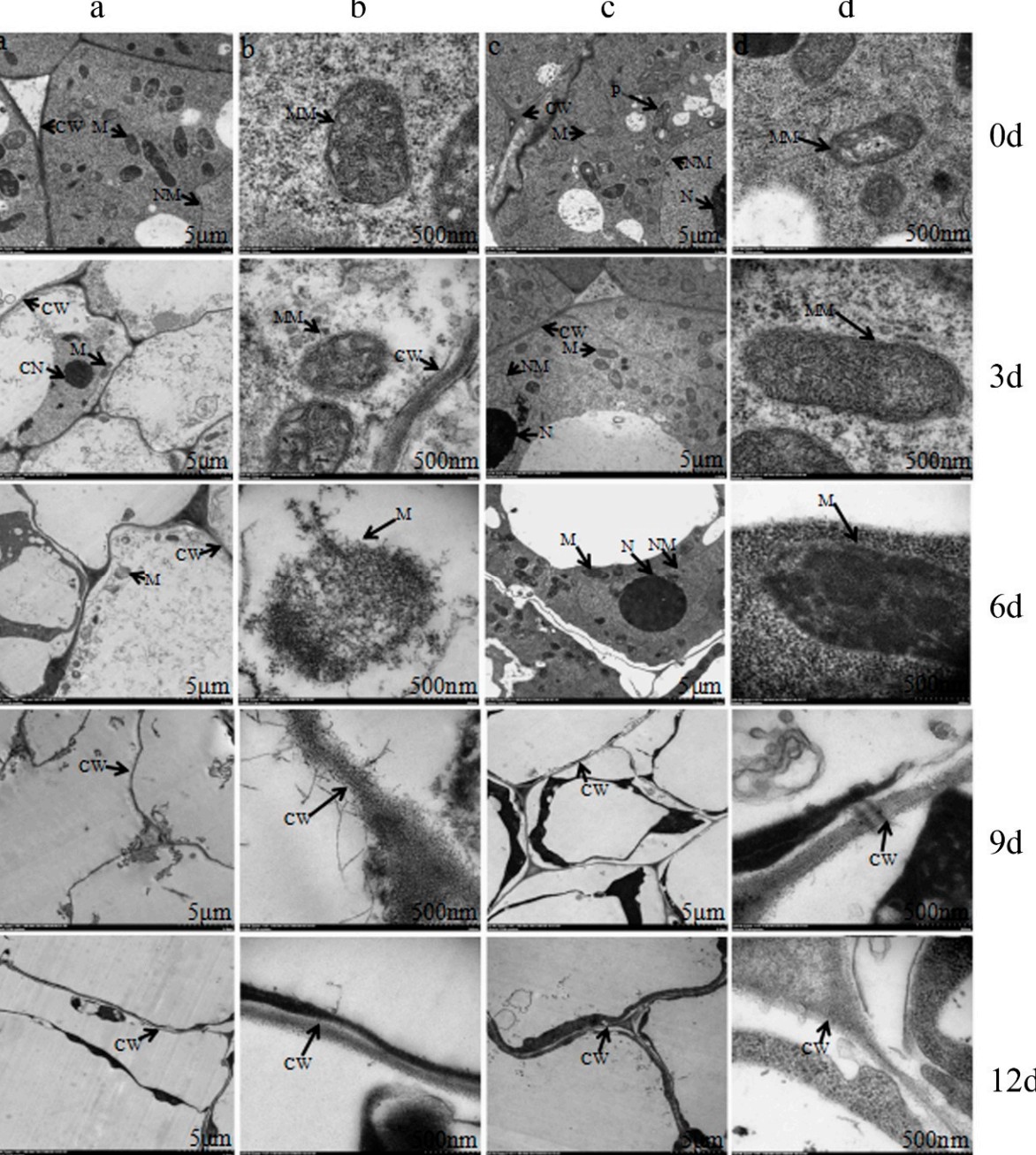

**Figure 3.** The effects of $Ca(NO_3)_2$ stress on the ultrastructure of mitochondria in roots of cucumber plants. (**a**,**b**) CQ and (**c**,**d**) BM. CW, cell wall; CN, cell nucleus; N, nucleolus; NM, nuclear membrane; M, mitochondrion; MM, mitochondrial membrane; P, plastid. The scale bars for root cells and mitochondria are 5 µm and 500 nm, respectively.

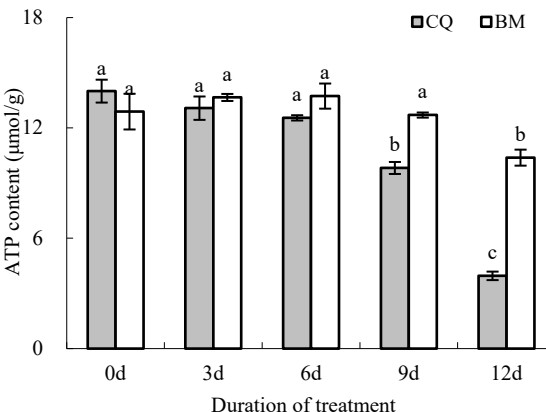

**Figure 4.** Determination of the ATP content in the mitochondria of cucumber roots at 0, 3, 6, 9, and 12 days after Ca(NO$_3$)$_2$ treatment. Values are the means of three independent experiments. Bars marked with dissimilar letters are significantly different according to Duncan's multiple range tests ($p < 0.05$).

### 3.5. Effects of Ca(NO$_3$)$_2$ on NO$_3^-$, NH$_4^+$ Content, and NR and NiR Enzyme Activity

NO$_3^-$ concentrations were much higher in plants under stress conditions (Figure 5A). After 0 and 3 days, Ca(NO$_3$)$_2$ stress did not cause a significant change in NO$_3^-$. However, after 6, 9, and 12 days, the nitrate concentrations levels increased conspicuously from the two varieties of cucumber and the nitrate concentrations were higher in BM than in CQ. Under Ca(NO$_3$)$_2$ stresses, the content of NH$_4^+$ first rose and then decreased (Figure 5B). After 3 days of treatment, NH$_4^+$ content of the two cucumber varieties began to decline. NR activity in the roots of the Ca(NO$_3$)$_2$-treated plants increased after 3 days and then began to decrease dramatically in the two cucumber varieties (Figure 5C). Meanwhile, NiR activity in the roots was significantly decreased by Ca(NO$_3$)$_2$ stress (Figure 5D).

### 3.6. NO Content and NOS Activity

The accumulation of NO in the two varieties of cucumber reached a peak after 9 days in Ca(NO$_3$)$_2$; however, NO content remained steady after 12 days. High levels of NO were detected in BM compared to CQ cucumber roots before 9 days (Figure 5E). The results showed NOS activity gradually declined after Ca(NO$_3$)$_2$ stress in the two cultivars at 0 days (Figure 5F). NOS activity in CQ remained lower compared to BM in all treatments.

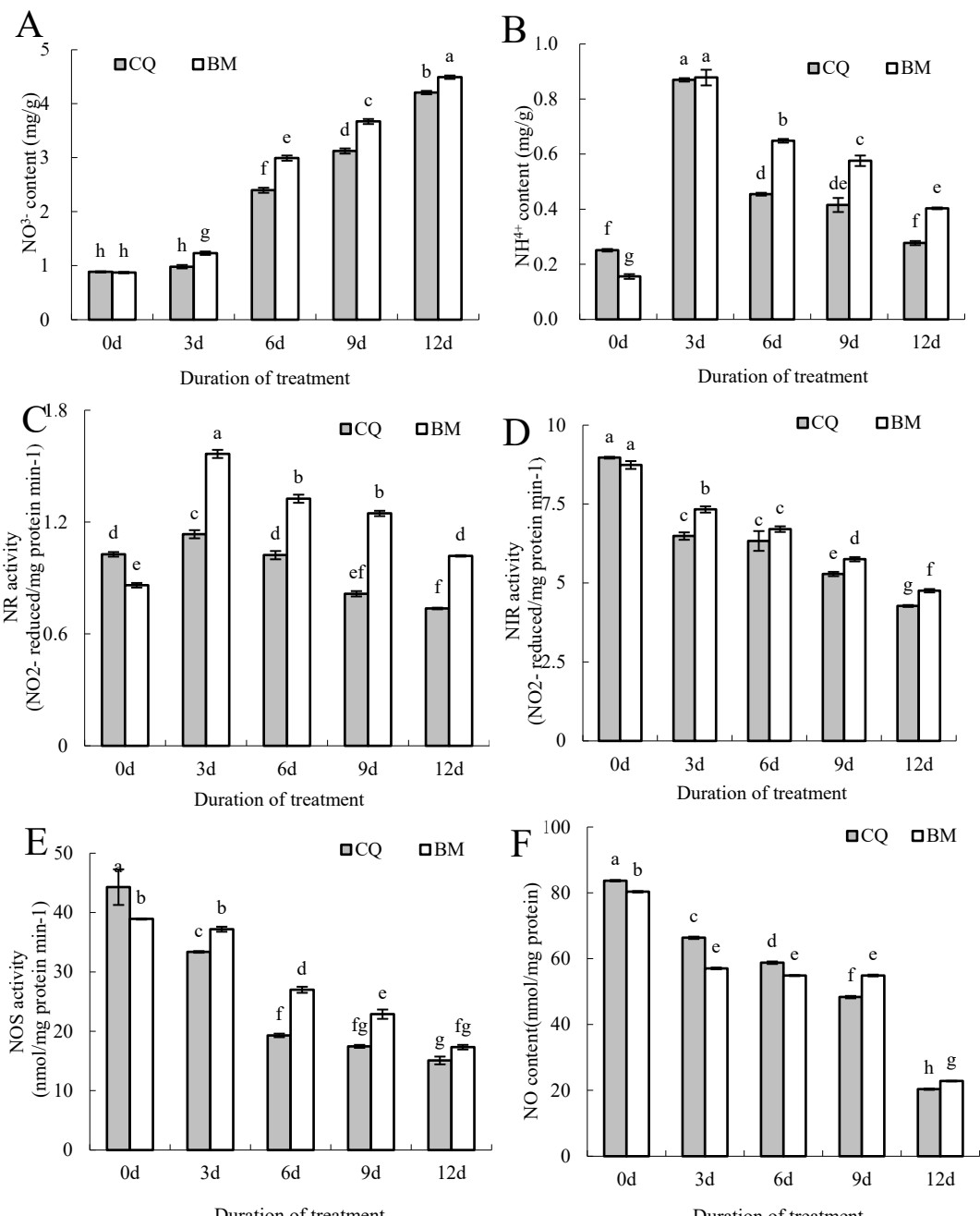

**Figure 5.** Determination of $NO_3^-$ content (**A**), $NH_4^+$ content (**B**), nitrite reductase (**C**), NIR activity (**D**), NOS activity (**E**), and NO content (**F**) in cucumber roots at 0, 3, 6, 9, and 12 days after $Ca(NO_3)_2$ treatment. Values are the means of three independent experiments. Bars marked with dissimilar letters are significantly different according to Duncan's multiple range tests ($p < 0.05$).

## 4. Discussion

In our present work, the effects of $Ca(NO_3)_2$ on cucumber root mitochondria were investigated in relation to nitrogen metabolism. In response to $Ca(NO_3)_2$, the mitochondrial numbers and structure of the two cultivars of cucumber showed great changes. This result could be explained through the osmotic stress effects of plant root growth [24]. Previous work showed that salt stress conditions had a suppression role in the growth of plants [25]. Similar results were obtained in our present study. We found that root growth parameters were significantly decreased by $Ca(NO_3)_2$ stress. It was also observed that $Ca(NO_3)_2$ treatments cause reduction in roots length, tips, and surface area, but the

average diameter was increased. Root morphology is related to the root cell's energy supply ability and the effective structure and enzyme activity of mitochondria [26]. In the presence of $Ca(NO_3)_2$, in order to maintain cell morphology and guarantee cell activity, cells usually increase organic solute to resist $Ca(NO_3)_2$ stress [27]. Increased organic solutes in cells cause mitochondrial membrane damage. However, harm to the mitochondrial membrane structure caused mitochondria response stress that was a production of ROS [28]. Our study showed that $Ca(NO_3)_2$ treatments increased ROS production and accumulation. There were continuous increases in ROS in both of the cultivar's root mitochondria, but the salt-tolerant cucumber was significantly delayed in its increase of MDA. Similar results were also found in grape [29]. Excess MDA and ROS lead to the mitochondrial antioxidant system clearing ROS. Our study showed that the MDA and ROS content in BM was lower than CQ after treatment. This indicates that BM's antioxidant system capacity is stronger than that of CQ. That is consistent with results of mitochondrial SOD, POD, and CAT activity in our study. Our study showed that the SOD and POD activity of BM were higher than that of CQ; furthermore, activities of SOD and POD decreased with the treatment [30]. This is consistent with results of research on resistance to high temperatures in Wucai (*Brassica campestris* L.) roots. However, CAT is playing the role of decomposing $H_2O_2$ that is converted to $H_2O$ by POD and CAT. $H_2O_2$ accumulated and increased the oxidative damage to the mitochondrial membrane [31].

Due to changes in mitochondria electron transport, electron leaks cause $O_2$ to produce $O_2^-$, along with changes in mitochondrial membrane potential. ROS generation increased when the electron transport chain (ETC) was blocked, resulting in dysfunctional carriers or production of ATP. Studies showed that rotenone inhibited electron flows and then enhanced ROS generation of complex I [32]. Meanwhile, complex II was also affected. Then, ATP production and the synthesis of sucrose were inhibited [33]. Similarly, complex III and IV also affected that balance of the mitochondrial electron transport chain and ATP production when they were inhibited. Leakage of electrons resulted in electrons from dehydrogenases being unable to reach the ubiquinone pool, and then $O_2$ in the mitochondria and electrons combined to form $O_2^-$ [34]. Complex II was not directly related to complex I, and the ubiquinone-generated electrons did not reach complex II. The role of complex II is similar to that of complex I, which provides electrons to the ubiquinone pool. When the activity of complex II was inhibited, not only was the electron transport chain blocked, but the process of the TCA cycle in which succinate dehydrogenase was involved was also inhibited [35]. As complex I activity decreased, electron transfer also decreased, which might lead to decreasing complex III and IV activities (as was observed in our study). At the same time, activities of complex III and complex IV were decreased, and $O_2$ in the mitochondrial matrix was converted to $O_2^-$. Production of ROS has an impact on mitochondrial membrane structure stability, such as the mitochondrial membrane permeability transition pore and mitochondrial membrane fluidity. Based on our results, we know that these two indicators are affected, consistent with the results of Douglas et al. who studied *Arabidopsis* protoplasts [36]. When the mitochondrial membrane structure is damaged, the amount of $NO_3^-$ infiltration gradually increased, which resulted in damage to mitochondria gradually increasing. Cytochrome c loosely binds to the mitochondrial inner membrane, whereas cytochrome a tightly binds to the mitochondrial inner membrane. When the mitochondrial membrane structure was destroyed, the cytochrome c/a ratio decreased due to the cytochromes dropping from the mitochondria intima.

Our results show that $Ca(NO_3)_2$ stress involves NO content change. The synthesis route of NO mainly consists of two parts. First, the process of NO produced by arginine is called the arginine pathway. Second, NO was generated by nitrogen metabolism. Our study results show that $NO_3^-$ content in plant roots increased under $Ca(NO_3)_2$ stress, and results were consistent with those of Zhen [37]. When the external $NO_3^-$ concentration is high, plant roots absorb some $NO_3^-$, resulting in increased $NO_3^-$ content in the root system [38]. As the $NO_3^-$ concentration increased, the enzyme that converts $NO_3^-$ to $NO_2^-$ was promoted. Results of our experiments show that NR activity increased first and then decreased with treatment, which was possibly due to feedback inhibition of higher concentrations of $NO_3^-$ [10]. However, the trend of NR activity changing in roots was the same among

the two cultivars, which indicated that the difference in salt tolerance had no significant effect on the trend of nitrogen metabolism. Our subsequent study also confirmed the above point of view. It is well-known that $NO_3^-$ is reduced to $NO_2^-$ by the action of NR and then decreased to $NH_4^+$ by NIR [39]. When NR activity is reduced, NIR cannot use the full substrate to catalyze the reaction, which results in the activity of NIR decreasing. Therefore, the content of $NH_4^+$ is slightly decreased, which is consistent with the results of our study under the stress of $Ca(NO_3)_2$. Many $NO_3^-$-converting processes involve several enzymes and nitric oxide synthase enzymes in the mitochondria, such as cytochrome c oxidase, aconitase, and succinate dehydrogenase, which were primarily engaged in the conversion of the $NO_2^-$ to NO effect [40]. Due to the decrease in NIR activity, the catalytic substrate of NOS has decreased and the activity of the enzyme is reduced. As a result, our study was the same as that described above.

In summary, our results strongly suggest that ROS accumulation plays an important role in root mitochondrial damage and plant root growth inhibition in cucumber seedlings under $Ca(NO_3)_2$ stress. These findings provide physiological insights into root damage in response to salt stress for salt tolerance in cucumber.

**Supplementary Materials:** The following are available online at http://www.mdpi.com/2073-4395/10/2/167/s1, Table S1: Effects of 80 mM $Ca(NO_3)_2$ Stress on Different Cucumber Cultivars, Table S2: Effects of Different $Ca(NO_3)_2$ Concentrations on Cucumber.

**Author Contributions:** Conceptualization, Y.Y. and Z.L.; methodology, J.L.; formal analysis, L.T.; validation, C.W., L.Y. and J.H.; software, S.J. and X.F.; resources, S.Z. All authors have read and agreed to the published version of the manuscript.

**Funding:** Anhui Province Key Research and Development Program (201904a06020057).

**Conflicts of Interest:** The authors declare that they have no conflict of interests.

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
