# Peer review of "Effects of Ca(NO3)2 Stress on Mitochondria and Nitrogen Metabolism in Roots of Cucumber Seedlings"

_agronomy, doi:10.3390/agronomy10020167_

Round 1

Reviewer 1 Report

The topic of the work presented is interesting, the manuscript however has numerous shortcomings and ambiguities. The two cucumber cultivars studied are claimed salt tolerant vs sensitive, but this statement is not substantiated nor any reference is given. Plants were apparently grown under no environmental control in a greenhouse in sand culture and hydroponics, which makes difficult to draw physiological conclusions from the results. Due to nitrate uptake of roots from hydroponics, alkalization of the medium is expected. This is not measured or mentioned, except a note at the end of Abstract and Discussion that alkaline stress tolerance was tested. In ultrastructural morphological observations any meaningful conclusion can only be deduced if source tissue of the cells is known. The text of the manuscript needs rewriting, the use of English language is poor. In some points serious mistakes are included in the text (e.g. POD is not producing, but consuming H2O2; line 232).
The topic of the manuscript is worth investigations, however serious improvements are needed both in methodology and interpretation of results.

Author Response

Reviewer #1:

Q (1): The two cucumber cultivars studied are claimed salt tolerant vs sensitive, but this statement is not substantiated nor any reference is given.

Response: Thank you for your good suggestions. The preliminary experiment was conducted to screen the tolerant cultivar and the sensitive one. Six cultivars were popular in domestic market of China. They are from Shenqing No. 1(marked SQ1; China, Shanghai, Shanghai Funong Seed Co.,Ltd.), Jinyou No. 35(marked JY; China, Tianjin, Tianjin Kernel vegetable Research Institute), Shenqing No.  5(marked SQ5; China, Shanghai, Shanghai Funong Seed Co.,Ltd.), BoXin No. 525(marked BX; China, Tianjin, Tianjin Derit Seed Co.,Ltd.), BoMei No. 49(marked BM; China, Tianjin, Tianjin Derit Seed Co.,Ltd.), and Chunqiu changjian(marked CQ; China, Beijing, Beijing Flower-goddess Agriculture Co.,Ltd). They were planted in the experiment basement of Anhui Agricultural University.

Seeds were disinfected with 55 °C warm water for 15 min, then washed thoroughly with deionised water. They were germinated in filter papers in petri dishes in the dark for approximately 24 h at 29 °C, When the seed radicle broke through the seed coat by about 2 mm, they were transferred into plastic nursery trays (50 × 50 × 5 cm) containing fine sand. The seedling temperature was controlled at 29 ± 2 °C in the day and 19 ± 2 °C in the night. After cotyledon expansion, seedings were cultivated with half-strength Hoagland nutrient solution; after true leaf expansion, seedings were planted with Hoagland nutrient solution. At the second-leaf stage, seedlings of uniform size were transferred to a crate that contained Hoagland nutrient solution and was aerated with an air pump at an interval of 20 min to maintain the dissolved oxygen (DO) level at 8.0 ± 0.2 mg l−1. Fifteen cucumber seedlings were planted in one crate. The seedlings were cultivated in a greenhouse. The highest temperature during the day was 31 °C, and the lowest temperature at night was 23 °C, and the relative humidity was 51%–71% during their growth period under natural illumination. After one day of the transplanting seedling stage, seedlings were treated with 80 mM Ca(NO3)2.  After 9 days, root morphology was measured by sampling, and the two cultivars with the greatest difference under stress were screened out according to root length and root tip number.

Table a. Effects of 80 mM Ca(NO3)2 Stress on Different Cucumber Cultivars

Length (cm)

SurfArea (cm2)

CQ

327.64 ± 3.55e

125.39 ± 1.32a

SQ1

331.36 ± 2.52e

89.73 ± 1.13e

JY

373.14 ± 1.37b

108.52 ± 0.59c

SQ5

356.51 ± 2.67c

105.56 ± 1.15c

BX

342.03 ± 1.06d

97.63 ± 0.54d

BM

388.25 ± 4.67a

119.33±1.21b

Note:Values represent the mean ± SE (n = 3). Letters indicate significant differences at P<0.05 according to Duncan’s multiple range tests.

According to table a, under the stress treatment of 80 mM Ca(NO3)2, the root length of 'CQ' was the smallest, the root length of 'BM' was the largest, and the surface area of 'CQ' and 'BM' was the largest. Root length of other varieties was in the middle, and the surface area of other varieties was lower than that of 'CQ' and 'BM'.

We will show it as a supplementary.

Please see lines 67-69

Q (2): Plants were apparently grown under no environmental control in a greenhouse in sand culture and hydroponics, which makes difficult to draw physiological conclusions from the results. Due to nitrate uptake of roots from hydroponics, alkalization of the medium is expected. This is not measured or mentioned, except a note at the end of Abstract and Discussion that alkaline stress tolerance was tested.

Response: Thank you for your good suggestions. We did a preliminary experiment. The two selected varieties were cultured in the same way, respectively treated with Ca(NO3)2 of 50, 60, 70, 80 and 90 mM. At 9 days, root morphology was measured by sampling, and the most significant stress concentration was screened out according to root length and root tip number.

Table b. Effects of Different Ca(NO3)2 Concentrations on Cucumber

Length (cm)

SurfArea (cm2)

CQ

BM

CQ

BM

0mM

541.55±3.43b

544.08±1.70b

173.34±1.50b

172.21±1.01b

50mM

612.25±5H.21a

623.65±3.04a

191.30±1.80a

188.88±2.41a

60mM

524.42±3.91c

544.01±2.72b

175.64±2.73b

167.14±2.08b

70mM

353.80±2.82d

414.61±1.58c

157.02±1.37c

126.27±1.84c

80mM

324.30±1.40e

388.25±1.71d

125.39±1.32d

118.67±2.28d

90mM

280.91±1.93f

332.60±1.34e

103.83±0.46e

116.3±0.463d

Note:Values represent the mean ± SE (n = 3). Letters indicate significant differences at P<0.05 according to Duncan’s multiple range tests.

According to table b, under the treatment of 60mm Ca(NO3)2, the length and surface area of 'CQ' and 'BM' increased, and both root length and surface area decreased with the increase of concentration. The change of length and surface area under 70 mM Ca(NO3)2 treatment was the biggest difference from that under 60 mM Ca(NO3)2 treatment, and the length and surface area gradually decreased with the increase of concentration.

We will show it as an supplementary

Please see the supplementary for details.

Q (3): In ultrastructural morphological observations any meaningful conclusion can only be deduced if source tissue of the cells is known.

Response: Thank you for your good suggestions. The source tissue of the cells is root tips.

Please see line 93.

Q (4): The text of the manuscript needs rewriting, the use of English language is poor. In some points serious mistakes are included in the text (e.g. POD is not producing, but consuming H2O2; line 232).

Response: Thank you for your good suggestions. We will use professional English editing services to ensure the accuracy of manuscript.

Please see lines 249-251.

Q (5): The topic of the manuscript is worth investigations, however serious improvements are needed both in methodology and interpretation of results.

Response: Thank you for your good suggestions. We have improved the Methods and interpretation of results.

Please see lines 67-84, 131-137,142-148, 154-158, 166-169, 203-205, 232-233, 238-240, 249-251.

Reviewer 2 Report

The manuscript presents an interesting and currently important research concerning the salinization of soil, which is one of the most important environmental factors that limit crop growth of plant.

However, I have many questions and comments

Not all abbreviations used are well described in the Materials and Methods e.g. abbreviations of the cultivars tested, etc.

Line 64: Is it about sensitive and tolerant cucumber cultivars? It is not clear.

How many seedlings transferred into plastic nursery trays? Have all seedlings been moved?

The seedlings were cultivated in greenhouse at 23-31°C and the relative humidity of 51-71% during their growth period under natural illumination. This description is too general. Under what conditions were the samples taken? Was it always at the same time? What was the photoperiod?

Line 73: What does mean 70mm Ca(NO3)2?

The design of the experiment is unclear.

Line 115: The data were statistically analyzed using one-way analysis of variance, and so the roots features, enzymes activity and also content other physiological parameters were analysed using one-way ANOVA. In the figures, compared data relate to two factors?: term of measurement and cultivar?

What exactly was the control?

The authors compare all results to treatment 0d. This is incorrect because the observations lasted 12 days and during this time e.g. seedlings' root length varied regardless of the salinity used, because the plants grew. Similarly with other features, the size of which depends on the age and stage of development of the tissues.

In the Results and Discussion, the authors also compare cultivars with each other, but they did not analyze it. There is a lack of statistical information about the significance of these relationships.

The work also lacks the relationship between the assessed morphometric and physiological and biochemical features under control and salinity conditions, e.g. PCA, correlation or regression analysis.

The idea is interesting, but the analysis of the results requires major changes.

Other remarks

Line 136: an error in the name H2O2

Line 280: in rice?

Author Response

Reviewer #2:

The manuscript presents an interesting and currently important research concerning the salinization of soil, which is one of the most important environmental factors that limit crop growth of plant. However, I have many questions and comments

Q (1): Not all abbreviations used are well described in the Materials and Methods e.g. abbreviations of the cultivars tested, etc.

Response: Thank you for your good suggestions. We have improved the Materials and Methods.

Please see lines 67-69.

Q (2): Line 64: Is it about sensitive and tolerant cucumber cultivars? It is not clear.

Response: Thank you for your good suggestions. The preliminary experiment was conducted to screen the tolerant cultivar and the sensitive one. Six cultivars were popular in domestic market of China. They are from Shenqing No. 1(marked SQ1; China, Shanghai, Shanghai Funong Seed Co.,Ltd.), Jinyou No. 35(marked JY; China, Tianjin, Tianjin Kernel vegetable Research Institute), Shenqing No.  5(marked SQ5; China, Shanghai, Shanghai Funong Seed Co.,Ltd.), BoXin No. 525(marked BX; China, Tianjin, Tianjin Derit Seed Co.,Ltd.), BoMei No. 49(marked BM; China, Tianjin, Tianjin Derit Seed Co.,Ltd.), and Chunqiu changjian(marked CQ; China, Beijing, Beijing Flower-goddess Agriculture Co.,Ltd). They were planted in the experiment basement of Anhui Agricultural University.

Seeds were disinfected with 55 °C warm water for 15 min, then washed thoroughly with deionised water. They were germinated in filter papers in petri dishes in the dark for approximately 24 h at 29 °C, When the seed radicle broke through the seed coat by about 2 mm, they were transferred into plastic nursery trays (50 × 50 × 5 cm) containing fine sand. The seedling temperature was controlled at 29 ± 2 °C in the day and 19 ± 2 °C in the night. After cotyledon expansion, seedings were cultivated with half-strength Hoagland nutrient solution; after true leaf expansion, seedings were planted with Hoagland nutrient solution. At the second-leaf stage, seedlings of uniform size were transferred to a crate that contained Hoagland nutrient solution and was aerated with an air pump at an interval of 20 min to maintain the dissolved oxygen (DO) level at 8.0 ± 0.2 mg l−1. Fifteen cucumber seedlings were planted in one crate. The seedlings were cultivated in a greenhouse. The highest temperature during the day was 31 °C, and the lowest temperature at night was 23 °C, and the relative humidity was 51%–71% during their growth period under natural illumination. After one day of the transplanting seedling stage, seedlings were treated with 80 mM Ca(NO3)2.  After 9 days, root morphology was measured by sampling, and the two cultivars with the greatest difference under stress were screened out according to root length and root tip number.

Table a. Effects of 80 mM Ca(NO3)2 Stress on Different Cucumber Cultivars

Length (cm)

SurfArea (cm2)

CQ

327.64 ± 3.55e

125.39 ± 1.32a

SQ1

331.36 ± 2.52e

89.73 ± 1.13e

JY

373.14 ± 1.37b

108.52 ± 0.59c

SQ5

356.51 ± 2.67c

105.56 ± 1.15c

BX

342.03 ± 1.06d

97.63 ± 0.54d

BM

388.25 ± 4.67a

119.33±1.21b

Note:Values represent the mean ± SE (n = 3). Letters indicate significant differences at P<0.05 according to Duncan’s multiple range tests.

According to table a, under the stress treatment of 80 mM Ca(NO3)2, the root length of 'CQ' was the smallest, the root length of 'BM' was the largest, and the surface area of 'CQ' and 'BM' was the largest. Root length of other varieties was in the middle, and the surface area of other varieties was lower than that of 'CQ' and 'BM'.

We will show it as a supplementary.

Please see lines 67-69

Q (3): How many seedlings transferred into plastic nursery trays? Have all seedlings been moved?

Response: Thank you for your good suggestions. Fifteen cucumber seedlings were planted in one crate.

Please see line 78

Q (4): The seedlings were cultivated in greenhouse at 23-31°C and the relative humidity of 51-71% during their growth period under natural illumination. This description is too general. Under what conditions were the samples taken? Was it always at the same time? What was the photoperiod?

Response: Thank you for your good suggestions. We have improved the Materials and Methods.

Please see lines 73-81.

Q (5): Line 73: What does mean 70mm Ca(NO3)2?

Response: Thank you for your good suggestions. The 70mM Ca(NO3)2 means that the most significant concentration of stress was screened. We did a preliminary experiment. The two selected varieties were cultured in the same way, respectively treated with Ca(NO3)2 of 50 mM, 60 mM, 70 mM, 80 mM and 90 mM. At 9 days, root morphology was measured by sampling, and the most significant stress concentration was screened out according to root length and root tip number.

Table b. Effects of Different Ca(NO3)2 Concentrations on Cucumber

Length (cm)

SurfArea (cm2)

CQ

BM

CQ

BM

0mM

541.55±3.43b

544.08±1.70b

173.34±1.50b

172.21±1.01b

50mM

612.25±5H.21a

623.65±3.04a

191.30±1.80a

188.88±2.41a

60mM

524.42±3.91c

544.01±2.72b

175.64±2.73b

167.14±2.08b

70mM

353.80±2.82d

414.61±1.58c

157.02±1.37c

126.27±1.84c

80mM

324.30±1.40e

388.25±1.71d

125.39±1.32d

118.67±2.28d

90mM

280.91±1.93f

332.60±1.34e

103.83±0.46e

116.3±0.463d

Note:Values represent the mean ± SE (n = 3). Letters indicate significant differences at P<0.05 according to Duncan’s multiple range tests.

According to table b, under the treatment of 60mm Ca(NO3)2, the length and surface area of 'CQ' and 'BM' increased, and both root length and surface area decreased with the increase of concentration. The change of length and surface area under 70 mM Ca(NO3)2 treatment was the biggest difference from that under 60 mM Ca(NO3)2 treatment, and the length and surface area gradually decreased with the increase of concentration.

We will show it as a supplementary

Please see line 82

Q (6): The design of the experiment is unclear. Line 115: The data were statistically analyzed using one-way analysis of variance, and so the roots features, enzymes activity and also content other physiological parameters were analysed using one-way ANOVA. In the figures, compared data relate to two factors?: term of measurement and cultivar?

Response: Thank you for your good suggestions. We redescribed the design of the experiment. We had used two-way ANOVA analysis and revised the contents of the manuscript.

Please see lines 67-84, 126-127.

Q (7): What exactly was the control? The authors compare all results to treatment 0d. This is incorrect because the observations lasted 12 days and during this time e.g. seedlings' root length varied regardless of the salinity used, because the plants grew. Similarly with other features, the size of which depends on the age and stage of development of the tissues.

Response: Thank you for your good suggestions. We have improved the methodology and interpretation of results.

Please see lines 67-84, 131-137,142-148, 154-158, 166-169.

Q (8): In the Results and Discussion, the authors also compare cultivars with each other, but they did not analyze it. There is a lack of statistical information about the significance of these relationships.

Response: Thank you for your good suggestions. We have improved the interpretation of results and Discussion.

Please see lines 126-132,137-143,149-153. 142-148, 154-158, 166-169, 203-205, 232-233, 238-240, 249-251.

Q (9): The work also lacks the relationship between the assessed morphometric and physiological and biochemical features under control and salinity conditions, e.g. PCA, correlation or regression analysis.

Response: Thank you for your good suggestions. We did a preliminary experiment. The two selected varieties were cultured in the same way, respectively treated with Ca(NO3)2 of 50, 60, 70, 80 and 90 mM. At 9 days, root morphology was measured by sampling, and the most significant stress concentration was screened out according to root length and root tip number.

Table b. Effects of Different Ca(NO3)2 Concentrations on Cucumber

Length (cm)

SurfArea (cm2)

CQ

BM

CQ

BM

0mM

541.55±3.43b

544.08±1.70b

173.34±1.50b

172.21±1.01b

50mM

612.25±5H.21a

623.65±3.04a

191.30±1.80a

188.88±2.41a

60mM

524.42±3.91c

544.01±2.72b

175.64±2.73b

167.14±2.08b

70mM

353.80±2.82d

414.61±1.58c

157.02±1.37c

126.27±1.84c

80mM

324.30±1.40e

388.25±1.71d

125.39±1.32d

118.67±2.28d

90mM

280.91±1.93f

332.60±1.34e

103.83±0.46e

116.3±0.463d

Note:Values represent the mean ± SE (n = 3). Letters indicate significant differences at P<0.05 according to Duncan’s multiple range tests.

According to table b, under the treatment of 60mm Ca(NO3)2, the length and surface area of 'CQ' and 'BM' increased, and both root length and surface area decreased with the increase of concentration. The change of length and surface area under 70 mM Ca(NO3)2 treatment was the biggest difference from that under 60 mM Ca(NO3)2 treatment, and the length and surface area gradually decreased with the increase of concentration.

We will show it as a supplementary

Please see line 66.

Q (10): The idea is interesting, but the analysis of the results requires major changes.

Response: Thank you for your good suggestions. We have improved the analysis of results. We will use professional English editing services to improve the accuracy of our manuscripts.

Please see lines 137-143,149-153. 142-148, 154-158, 166-169, 203-205, 232-233, 238-240, 249-251.

Q (11): Other remarks

Line 136: an error in the name H2O2

Line 280: in rice?

Response: Thank you for your good suggestions. We will use professional English editing services to improve the accuracy of our manuscripts.

Please see lines 149, 297-298.

Reviewer 3 Report

Interesting work, deals with the topic of the impact of abiotic stres on cucumber plants in the early stages of development - seedlings.

A lot of work has been done in relation to morphological parameters and biochemical reactions of plants under alkaline stress conditions. In the work there is a lack of connection of these elements for a better understanding the relationship between all assessed parameters. PCA analysis or other form would be helpful - for example correlation relationship.

My questions and comments

Materials and Methods. Abbreviations used and described in the are not applied to all factors, e.g. abbreviations of the cultivars tested – they are used later in tables and figures (but not in this part described).

Line 64: sensitive and tolerant cucumber cultivars – should be add “variety” to for clear.

Seedlings transferred into plastic nursery trays – how many seedlings?

“The seedlings were cultivated in greenhouse at 23-31°C and the relative humidity of 51-71% during their growth period under natural illumination” – it needs more information – duration of day/night and the temperature during these periods as well as humidity (for each separately), because differences between them are big.

There is no information if some of the plants were kept in conditions of Hoagland nutrient solution and the others in stress condition during the same time. If only in stress conditions there is no control according to particular term of observations and the authors can compare values of parameters only up to 0 day in conditions of stress, and yet the plants develop and their vital parameters change. Please describe it in more detail.

Line 73: What does mean 70mm Ca(NO3)2? What was the concentration of the solution?

The design of the experiment is unclear.

Line 115: The data were statistically analyzed using one-way analysis of variance, and so the roots features, enzymes activity and also content other physiological parameters were analysed using one-way ANOVA.

Table 1. according to morphological structure of roots it is true (the comparison for each trait is separate for each variety). But the control is 0 day, when the seedlings were developed differently than on 3, 6, 9 and 12 days. The same to Table 2.

In the figures, compared data relate to two factors: term of measurement and cultivar (using two-way ANOVA, as I suppose. Because of that Authors should describe “Statistical analysis” properly.

In the Results and Discussion, the authors compare cultivars with each other, but they did not analyze it and there is a lack of statistical information about the significance of these relationships.

I suggest also to show the relationship between the assessed morphometric and physiological and biochemical features under control and stress conditions, e.g. correlation or regression or PCA analysis.

The idea is interesting, but the analysis of the results requires major changes.

In summary -  the last sentence is not understood: .........................in response to alkaline stress for alkaline tolerance in cucumber.

English throughout the work should be checked and corrected, that the content was clear and grammatically correct.

Other remarks:

Line 136: hydrogen peroxide H2O2 (not peroxide)

Line 280: in rice seedlings under........?

Author Response

Reviewer #3:

Q (1): Materials and Methods. Abbreviations used and described in the are not applied to all factors, e.g. abbreviations of the cultivars tested – they are used later in tables and figures (but not in this part described).

Response: Thank you for your good suggestions. We have improved the Materials and Methods.

Please see lines 67-84.

Q (2): Line 64: sensitive and tolerant cucumber cultivars – should be add “variety” to for clear.

Response: Thank you for your good suggestions. The preliminary experiment was conducted to screen the tolerant cultivar and the sensitive one. Six cultivars were popular in domestic market of China. They are from Shenqing No. 1(marked SQ1; China, Shanghai, Shanghai Funong Seed Co.,Ltd.), Jinyou No. 35(marked JY; China, Tianjin, Tianjin Kernel vegetable Research Institute), Shenqing No.  5(marked SQ5; China, Shanghai, Shanghai Funong Seed Co.,Ltd.), BoXin No. 525(marked BX; China, Tianjin, Tianjin Derit Seed Co.,Ltd.), BoMei No. 49(marked BM; China, Tianjin, Tianjin Derit Seed Co.,Ltd.), and Chunqiu changjian(marked CQ; China, Beijing, Beijing Flower-goddess Agriculture Co.,Ltd). They were planted in the experiment basement of Anhui Agricultural University.

Seeds were disinfected with 55 °C warm water for 15 min, then washed thoroughly with deionised water. They were germinated in filter papers in petri dishes in the dark for approximately 24 h at 29 °C, When the seed radicle broke through the seed coat by about 2 mm, they were transferred into plastic nursery trays (50 × 50 × 5 cm) containing fine sand. The seedling temperature was controlled at 29 ± 2 °C in the day and 19 ± 2 °C in the night. After cotyledon expansion, seedings were cultivated with half-strength Hoagland nutrient solution; after true leaf expansion, seedings were planted with Hoagland nutrient solution. At the second-leaf stage, seedlings of uniform size were transferred to a crate that contained Hoagland nutrient solution and was aerated with an air pump at an interval of 20 min to maintain the dissolved oxygen (DO) level at 8.0 ± 0.2 mg l−1. Fifteen cucumber seedlings were planted in one crate. The seedlings were cultivated in a greenhouse. The highest temperature during the day was 31 °C, and the lowest temperature at night was 23 °C, and the relative humidity was 51%–71% during their growth period under natural illumination. After one day of the transplanting seedling stage, seedlings were treated with 80 mM Ca(NO3)2.  After 9 days, root morphology was measured by sampling, and the two cultivars with the greatest difference under stress were screened out according to root length and root tip number.

Table a. Effects of 80 mM Ca(NO3)2 Stress on Different Cucumber Cultivars

Length (cm)

SurfArea (cm2)

CQ

327.64 ± 3.55e

125.39 ± 1.32a

SQ1

331.36 ± 2.52e

89.73 ± 1.13e

JY

373.14 ± 1.37b

108.52 ± 0.59c

SQ5

356.51 ± 2.67c

105.56 ± 1.15c

BX

342.03 ± 1.06d

97.63 ± 0.54d

BM

388.25 ± 4.67a

119.33±1.21b

Note:Values represent the mean ± SE (n = 3). Letters indicate significant differences at P<0.05 according to Duncan’s multiple range tests.

According to table a, under the stress treatment of 80 mM Ca(NO3)2, the root length of 'CQ' was the smallest, the root length of 'BM' was the largest, and the surface area of 'CQ' and 'BM' was the largest. Root length of other varieties was in the middle, and the surface area of other varieties was lower than that of 'CQ' and 'BM'.

We will show it as a supplementary.

Please see lines 67-69.

Q (3): Seedlings transferred into plastic nursery trays – how many seedlings?

Response: Thank you for your good suggestions. Fifteen cucumber seedlings were planted in one crate.

Please see line 78.

Q (4): “The seedlings were cultivated in greenhouse at 23-31°C and the relative humidity of 51-71% during their growth period under natural illumination” – it needs more information – duration of day/night and the temperature during these periods as well as humidity (for each separately), because differences between them are big.

Response: Thank you for your good suggestions. We have improved the Materials and Methods.

Please see lines 70-81.

Q (5): There is no information if some of the plants were kept in conditions of Hoagland nutrient solution and the others in stress condition during the same time. If only in stress conditions there is no control according to particular term of observations and the authors can compare values of parameters only up to 0 day in conditions of stress, and yet the plants develop and their vital parameters change. Please describe it in more detail.

Response: Thank you for your good suggestions. The plants were kept in conditions of Hoagland nutrient solution and we calculated the Ca(NO3)2 concentration to be 70mM .We have improved the methodology and interpretation of results.

Please see lines 67-84, 131-137,142-148, 154-158, 166-169.

Q (6): Line 73: What does mean 70mm Ca(NO3)2? What was the concentration of the solution?

Response: Thank you for your good suggestions. It should be 70mM Ca(NO3)2. We did a preliminary experiment. The two selected varieties were cultured in the same way, respectively treated with Ca(NO3)2 of 50 , 60 , 70 , 80 and 90 mM. At 9 days, root morphology was measured by sampling, and the most significant stress concentration was screened out according to root length and root tip number.

Table b. Effects of Different Ca(NO3)2 Concentrations on Cucumber

Length (cm)

SurfArea (cm2)

CQ

BM

CQ

BM

0mM

541.55±3.43b

544.08±1.70b

173.34±1.50b

172.21±1.01b

50mM

612.25±5H.21a

623.65±3.04a

191.30±1.80a

188.88±2.41a

60mM

524.42±3.91c

544.01±2.72b

175.64±2.73b

167.14±2.08b

70mM

353.80±2.82d

414.61±1.58c

157.02±1.37c

126.27±1.84c

80mM

324.30±1.40e

388.25±1.71d

125.39±1.32d

118.67±2.28d

90mM

280.91±1.93f

332.60±1.34e

103.83±0.46e

116.3±0.463d

Note:Values represent the mean ± SE (n = 3). Letters indicate significant differences at P<0.05 according to Duncan’s multiple range tests.

According to table b, under the treatment of 60mm Ca(NO3)2, the length and surface area of 'CQ' and 'BM' increased, and both root length and surface area decreased with the increase of concentration. The change of length and surface area under 70 mM Ca(NO3)2 treatment was the biggest difference from that under 60 mM Ca(NO3)2 treatment, and the length and surface area gradually decreased with the increase of concentration.

We will show it as a supplementary

Please see line 82

Q (7): The design of the experiment is unclear.

Line 115: The data were statistically analyzed using one-way analysis of variance, and so the roots features, enzymes activity and also content other physiological parameters were analysed using one-way ANOVA.

Response: Thank you for your good suggestions. We redescribed the design of the experiment. We had used two-way ANOVA analysis and revised the contents of the manuscript.

Please see lines 67-84, 126-127.

Q (8): Table 1. according to morphological structure of roots it is true (the comparison for each trait is separate for each variety). But the control is 0 day, when the seedlings were developed differently than on 3, 6, 9 and 12 days. The same to Table 2.

Response: Thank you for your good suggestions. We modified the description.

Please see lines 131-137,142-148, 154-158, 166-169.

Q (9): In the figures, compared data relate to two factors: term of measurement and cultivar (using two-way ANOVA, as I suppose. Because of that Authors should describe “Statistical analysis” properly.

Response: Thank you for your good suggestions. We have used two-way ANOVA analysis and revised the contents of the manuscript.

Please see the whole manuscript.

Q (10): In the Results and Discussion, the authors compare cultivars with each other, but they did not analyze it and there is a lack of statistical information about the significance of these relationships.

I suggest also to show the relationship between the assessed morphometric and physiological and biochemical features under control and stress conditions, e.g. correlation or regression or PCA analysis.

The idea is interesting, but the analysis of the results requires major changes.

Response: Thank you for your good suggestions. We re-described the analysis of the results.

Please see lines 137-143,149-153. 142-148, 154-158, 166-169, 203-205, 232-233, 238-240, 249-251.

Q (11): In summary - the last sentence is not understood: .........................in response to alkaline stress for alkaline tolerance in cucumber.

Response: Thank you for your good suggestions. We modified the sentence.

Please see lines 299-300.

Q (12): English throughout the work should be checked and corrected, that the content was clear and grammatically correct.

Response: Thank you for your good suggestions. We will use professional English editing services to improve the accuracy of our manuscripts.

Q (13): Other remarks:

Line 136: hydrogen peroxide H2O2 (not peroxide)

Line 280: in rice seedlings under........?

Response: Thank you for your good suggestions. We will use professional English editing services to improve the accuracy of our manuscripts.

Please see lines 149,297-298.

Round 2

Reviewer 2 Report

In studies, the authors showed how 70 mM Ca (NO3)2 salinity affects:

- morphological features of plant roots,

- enzymes involved in mitochondria antioxidant system of seedlings,

- mitochondrial electron transfer chain

- NO metabolism.

However, it was not shown whether the changes in the values of the tested parameters due to salinity are significant compared to those obtained by plants under control conditions (without salinity).

Line 85: The root of the control and treated seedlings was sampled 0, 3, 6, 9, and 12 days after treatment.

Why were control samples taken? For what purpose?

There is no reference in the paper to the results obtained under control conditions.

There is a lack of presentation of control results and also their comparison with results obtained in salinity conditions. Perhaps showing relationships (e.g. correlations) between individual traits under control and salinity stress would show a complete analysis of results and their changes. Supplementary information shows only part of the results and explains the issues related to the morphometric features of the roots. And what about enzymes involved in mitochondria antioxidant system, and changes in mitochondrial electron transfer chain and NO metabolism?

Other remarks:

Quality of Tables should be improved.

Fig 4. (B, F) letters on the bars should be checked.

Line 173-177 no Figure.....

Links to figures and tables should be checked in the text, e.g.: Line 185 Fig 6A? There is no such figure.

Author Response

Reviewer #2:

Q (1): In studies, the authors showed how 70 mM Ca (NO3)2 salinity affects:

- morphological features of plant roots,

- enzymes involved in mitochondria antioxidant system of seedlings,

- mitochondrial electron transfer chain

- NO metabolism.

Response: Thank you for your good suggestions. In this study we investigated salt-sensitive Chunqiu (CQ) and salt-tolerant BoMei 49 (BM) seedlings, changes to enzymes involved in mitochondria antioxidant system seedlings, and changes in MPTP opening, mitochondrial membrane fluidity, mitochondrial membrane potential depolarization, mitochondrial electron transfer chain and NO metabolism in response to Ca(NO3)2. Metabolism is the focus of our follow-up research.

Please see lines 17-18.

Q (2): However, it was not shown whether the changes in the values of the tested parameters due to salinity are significant compared to those obtained by plants under control conditions (without salinity).

Response: Thank you for your good suggestions. In this study, two cucumber seedlings were treated with 70mM Ca(NO3)2 and sampled (roots) at 0,3,6,9 and 12 days to measure the relevant indexes(changes to enzymes involved in mitochondria antioxidant system seedlings, and changes in MPTP opening, mitochondrial membrane fluidity, mitochondrial membrane potential depolarization, mitochondrial electron transfer chain and NO metabolism).

Q (3): Line 85: The root of the control and treated seedlings was sampled 0, 3, 6, 9, and 12 days after treatment.

Why were control samples taken? For what purpose?

There is no reference in the paper to the results obtained under control conditions.

There is a lack of presentation of control results and also their comparison with results obtained in salinity conditions. Perhaps showing relationships (e.g. correlations) between individual traits under control and salinity stress would show a complete analysis of results and their changes. Supplementary information shows only part of the results and explains the issues related to the morphometric features of the roots. And what about enzymes involved in mitochondria antioxidant system, and changes in mitochondrial electron transfer chain and NO metabolism?

Response: Thank you for your good suggestions. In this study, six cucumber varieties were tested for root-related indexes after Ca(NO3)2 stress, and salt-sensitive Chunqiu (CQ) and salt-tolerant BoMei 49 (BM) were selected as test materials. We compared the changes of two cucumber varieties (salt-sensitive ‘CQ’ and salt-tolerant ‘BM’) on mitochondria and nitrogen metabolism in roots of cucumber seedlings at different periods of 0, 3, 6, 9, and 12 days under 70 mM Ca (NO3) 2 stress. The root of the treated seedlings was sampled 0, 3, 6, 9, and 12 days after treatment.

Please see line 86.

Q (4): Other remarks:

Quality of Tables should be improved.

Fig 4. (B, F) letters on the bars should be checked.

Line 173-177 no Figure.....

Links to figures and tables should be checked in the text, e.g.: Line 185 Fig 6A? There is no such figure.

Response: Thank you for your good suggestions. We recognized the problem and corrected it.

Please see lines 214-215,172-173,186.
